# Postoperative Changes in Body Composition Predict Long-Term Prognosis in Patients with Gastric Cancer

**DOI:** 10.3390/cancers17050738

**Published:** 2025-02-21

**Authors:** Kiyohiko Shuto, Yoshihiro Nabeya, Mikito Mori, Masato Yamazaki, Chihiro Kosugi, Kazuo Narushima, Akihiro Usui, Hiroyuki Nojima, Hiroaki Shimizu, Keiji Koda

**Affiliations:** 1Department of Surgery, Teikyo Chiba Medical Center, 3426-3 Anesaki, Ichihara-shi, Chiba 299-0111, Japan; mikimori45@yahoo.co.jp (M.M.); ckosugi0126@yahoo.co.jp (C.K.); ausui@med.teikyo-u.ac.jp (A.U.); h-iro@qg7.so-net.ne.jp (H.N.); h-shimizu@med.teikyo-u.ac.jp (H.S.); k-koda@med.teikyo-u.ac.jp (K.K.); 2Division of Esophago-Gastrointestinal Surgery, Chiba Cancer Center, Nitona-cho, Chiba 260-8717, Japan

**Keywords:** gastric cancer, computed tomography, skeletal muscle, body fat, prognosis

## Abstract

This study evaluated the changes in the quantity and quality of skeletal muscle and the quantity of body fat after radical gastrectomy for gastric cancer patients using computed tomography (CT) and investigated the relationship between body composition parameters and long-term prognosis. As a result, postoperative reduction and hyperintensity of skeletal muscle and body fat reduction were prognostic factors. Body composition scores using these three parameters predicted the prognosis of postoperative patients with gastric cancer well.

## 1. Introduction

Gastric cancer (GC) is the third most common cause of cancer-related death in Japan [1]. With the exception of some endoscopically resectable T1 cancers, surgical resection with lymphadenectomy is the most powerful treatment for GC [2]. However, after gastrectomy, patients are likely to have decreased digestive capacity and food intake, as well as weight loss, with subsequent malnutrition [3]. In patients with cancer, muscle and fat loss associated with nutritional disorders reduce activities of daily living (ADL) and have a negative impact on postoperative survival. Various parameters such as body mass index (BMI), skeletal muscle volume, subcutaneous fat volume, and visceral fat volume have been proposed as body composition (BC) biomarkers [4,5]. BMI is a simplified nutritional parameter available for routine assessment; however, obesity with increased BMI and decreased skeletal muscle volume, known as sarcopenic obesity, is also considered to have a negative impact on life outcomes [6]. BMI does not discriminate between body fat mass and lean body mass, and there is limited evidence to support that BMI is a powerful prognostic factor in cancer patients [7].

Because computed tomography (CT) is an objective modality to assess skeletal muscle and adipose tissue separately and is routinely used in clinical practice, skeletal muscle volume and body fat volume evaluated by CT was recently employed as a more detailed nutritional parameter [8]. When limited to surgical cases with GC, myopenia, a status of reduced skeletal muscle quantity, increases postoperative complications and worsens long-term prognosis, and myosteatosis, a condition of impaired skeletal muscle quality with decreased muscle density due to intramuscular fat infiltration, prolongs surgery time and increases postoperative complications [5,8,9,10].

Although many studies have examined the influence of BCs on surgical outcome after gastrectomy, most of these focused on preoperative patient status [4,5,6,8,9,10,11]. Few documents have referred to postoperative assessments of BC [7,12,13,14]. Furthermore, few studies have comprehensively investigated postoperative changes in these BC parameters such as skeletal muscle volume, skeletal muscle density, and body fat tissue [7]. It remains uncertain how much these BC parameters changes during the post-gastrectomy period and how much prognostic impact postoperative BC changes may have. Therefore, in this study, we investigated the impact of postoperative comprehensive changes in skeletal muscle and body fat quantity and skeletal muscle quality on long-term prognosis in patients with GC.

## 2. Materials and Methods

### 2.1. Patients

The study was carried out following the Helsinki Declaration and received approval from the institutional review board at the Teikyo University Chiba Medical Center (ethical committee approval No.18-171). From December 2005 to July 2023, the medical records of patients with gastric adenocarcinoma who underwent gastrectomy at Department of Surgery, Teikyo University Chiba Medical Center were investigated. A total of 540 patients were reviewed, of which 366 were enrolled in this study according to the following exclusion criteria: non-curative R1/R2 resection (*n* = 47); survival, death, or relapse within six months postoperatively (*n* = 36); use of home nutritional supplementation after discharge (*n* = 31); synchronous double cancer (*n* = 18); gastrectomy with thoracotomy approach (*n* = 15); pancreaticoduodenectomy (*n* = 4); and data deficiency (*n* = 23).

### 2.2. Treatment and Follow-Up

Treatment and follow-up were planned according to the treatment guidelines in Japan at that time [2]. Prior to surgery, patients underwent preoperative examinations that included physical examination, blood tests, gastroduodenoscopy, CT scan, and upper gastrointestinal series. R0 gastrectomy with lymphadenectomy was performed. Reduced extent of lymphadenectomy was selected for patients in insufficient general condition. Pathological findings were translated according to the 8th edition of the tumor–node–metastasis classification of the Union for International Cancer Control [15]. The prevalence of preoperative comorbidities was estimated using the updated Charlson comorbidity index [16], and the severity of postoperative complications was determined using the Clavien–Dindo classification system [17]. Neoadjuvant chemotherapy with S-1 + oxaliplatin was allocated to recent patients with cStage III tumors (*n* = 22). Postoperative adjuvant chemotherapy was administered for 1 year after surgery in patients with pStage II or above, as follows: S-1 alone for pStage II/III (*n* = 131); S-1 + cisplatin or oxaliplatin for pStage III/IV (*n* = 18); or S-1+ docetaxel for pStage III (*n* = 4).

During the postoperative follow-up course, blood tests were performed every three months, CT scans every six months, and gastroduodenoscopy every year for five years to monitor for recurrence. Post-discharge daily physical exercise was entrusted to the patient. Patients with recurrence received chemotherapy: first-line with S-1 + cisplatin or oxaliplatin; second-line with paclitaxel + ramucirumab; and third-line with nivolumab or irinotecan, if possible. The median postoperative follow-up time was 2344 days (interquartile range [IQR], 1295−3757). A total of six patients (1.6%) were lost to follow-up during the 5-year postoperative course.

### 2.3. Measurements of BC Parameters

CT scanning was performed within one month prior to surgery using a 64-row multidetector CT scanner (Lightspeed VCT, Revolution EVO; GE Healthcare, Wauwatosa, WI, USA). A non-enhanced scan of the upper-middle abdomen was obtained during a breath hold with the following parameters: 120 kVp tube voltage, 200−600 mA tube current, 1.0−1.5 spiral pitch, and 5-mm reconstruction interval. After completion of the non-enhanced scan, a routine contrast-enhanced thoraco-abdominal study was performed administering non-ionic iodinated contrast agents, unless contraindicated. Image data were transferred to an image processing workstation. Two experienced gastrointestinal imaging researchers (M.M. and K.N.) analyzed the cross-sectional non-enhanced CT image at the third lumbar vertebra level to measure the following three BC parameters: skeletal muscle volume (cm^2^), muscle density (Hounsfield unit [HU]), and body fat volume (cm^2^). The right and the left psoas major muscles were each semi-automatically traced, and total volume and total CT density were measured (Figure 1a). Body fat volume was determined by the sum of subcutaneous adipose tissue area and visceral adipose tissue area with tissue thresholds of −190 to −30 HU [4] using a commercially based software (Fat tissue analyzer version 3.0, Virtual Place Fujin Raijin; Aze, Canon Medical Systems, Tochigi, Japan) (Figure 1b). Postoperative non-enhanced CT images were obtained six months after surgery (median, 184 days; IQR, 169−208 days), and BC parameters were estimated with the same method. Volume data were adjusted by patient height (cm^2^/m^2^). Changes in muscle and fat volumes were assessed by percentages ((postoperative value)/(preoperative value) × 100%), and muscle density change by difference ((postoperative value)—(preoperative value) HU).

### 2.4. Cutoff Values of BC Parameter Changes

Sex-adjusted cut-off values of BC parameters were used for evaluation. Receiver operating characteristic (ROC) analysis for survival status during the follow-up period was performed to determine the optimal cutoff values by calculating the Youden index. As a result, rounded cutoff values for BC changes in muscle volume, fat volume, and muscle density were adopted as follows: male, 90%, 60%, and +12 HU, respectively; and female, 90%, 70%, and +12 HU, respectively.

### 2.5. Statistical Analysis

Continuous variables were expressed as median [IQR]. Survival curves were estimated by the Kaplan–Meier method to calculate cumulative survival rates. The Cox proportional hazards model was used for prognostic analyses to assess hazard ratios (HRs) with 95% confidence intervals (CI), and logistic regression was applied for analysis of risk factors to estimate odds ratios (ORs). A *p*-value of <0.05 was considered statistically significant. Significant variables achieved in the univariate analysis were inputted into a subsequent multivariate analysis. In prognostic analyses, cutoff values for continuous variables were determined by calculating the Youden index in the ROC analysis for survival status. In risk factor analyses for BC change, categorical variables were classified according to the minimum *p*-value in the chi-square test for BC change status, and continuous variables were divided by calculating the Youden index. All tests were conducted using SPSS software (version 28; IBM Corp., Armonk, NY, USA).

## 3. Results

### 3.1. Patient Characteristics

The characteristics of the patients are summarized in Table 1. Stage IV tumors were observed in five patients (1%): four patients received concurrent hepatectomy and one underwent adrenalectomy. Preoperative median BC parameters were as follows: muscle volume, 5.05 cm^2^/m^2^; body fat volume, 85.4 cm^2^/m^2^; and muscle density, 81.7 HU. The respective median BC changes at six months postoperatively were 94.2%, 60.6%, and +7.1 HU, with marked body fat loss and muscle density increase. Regarding correlations between BC parameters, almost no or very weak correlation was observed between muscle volume changes and fat volume changes (*r* = 0.201), a negative correlation between fat volume changes and muscle density changes (*r* = −0.430), and no correlation between muscle volume changes and muscle density changes (*r* = 0.110) (Figure 2).

### 3.2. Prognosis According to BC Parameters

Based on the sex-adjusted cutoffs as described above, patients were classified according to respective BC parameter changes as follows: muscle volume decrease group vs. non-decrease (*n* = 114 vs. *n* = 252), fat volume decrease vs. non-decrease (*n* = 195 vs. *n* = 171), and muscle density increase vs. non-increase (*n* = 119 vs. *n* = 247). Survival curves for each parameter are depicted in Figure 3 and demonstrate significantly worse 5-year overall survival (5y-OS) rate in the decreased muscle volume (5y-OS, 49% vs. 86%; HR, 4.687 (3.025–7.262), *p* < 0.001), decreased fat volume (5y-OS, 70% vs. 82%; HR, 1.782 (1.138–2.790), *p* = 0.012), and increased muscle density (5y-OS, 64% vs. 81%; HR, 2.130 (1.392–3.260), *p* < 0.001) groups.

### 3.3. Prognostic Risk Scoring According to BC Changes

We evaluated comprehensive BC changes and patient prognosis with a risk scoring method by substituting each HR (muscle volume, HR = 4.687; fat volume, HR = 1.782; and muscle density, HR = 2.130) for a rounded ratio of 2:1:1. The risk scores were then determined as follows: muscle volume decrease, 2 points; fat volume decrease, 1 point; and muscle density increase, 1 point. When all patients were reevaluated using these scores, they were divided into the following five minor subgroups: score 0, *n* = 105 (28%); score 1, *n* = 81 (22%); score 2, *n* = 103 (28%); score 3, *n* = 53 (15%); and score 4, *n* = 24 (7%). 5y-OS became worse depending on the score: score 0, 94%; score 1, 82%; score 2, 73%; score 3, 56%; and score 4, 20% (Figure 4a). Compared with the score 0 group (HR, 1), the respective HRs were: score 1, 3.074 (1.181–7.999); score 2, 4.789(1.971–11.635); score 3, 9.364 (3.777–23.214); and score 4, 23.852 (9.433–60.309). For further evaluation, minor five subgroups from score 0 to 4 were divided into two major subgroups. Because the survival curve of the entire patient cohort was located between those of the score 1 and score 2 subgroups, with a 5y-OS of 75%, patients with score 0 or score 1 were assigned to the BC non-changes group (*n* = 186), and patients with score 2 or more to the BC changes group (*n* = 180). In accordance with this major classification, the BC changes group had a significantly poor prognosis (5y-OS, 61% vs. 89%; HR, 4.049 (2.452–6.687), *p* < 0.001) (Figure 4b).

### 3.4. Prognostic Significance of BC Changes

Table 2 shows the results of the prognostic analysis for survival. As a result of the multivariate analysis, BC change (score > 2) was identified as an independent poor prognosticator (HR, 3.086 (1.831–5.202), *p* < 0.001), as well as older age (HR, 2.370 (1.506–3.729), *p* < 0.001), higher Charlson comorbidity index (HR, 1.799 [1.116–2.900], *p* = 0.016), total gastrectomy (HR, 1.828 (1.136–2.942), *p* = 0.013), and advanced pathological stage (HR, 2.377 (1.256–4.501), *p* = 0.008). Preoperative lower BMI was considered nearly an independent factor (HR, 1.594 (1.000–2.539), *p* = 0.050). Period of surgery was not identified as a prognostic factor.

Figure 5 depicts OS and disease-specific survival (DSS) according to cancer progression. Among patients with pathological stage I cancer, OS was significantly worse in the BC changes group compared with those of the non-changes group (73% vs. 94%, *p* < 0.001), whereas DSS was statistically similar (95% vs. 98%, *p* = 0.201) (Figure 5a,b). Meanwhile, in patients with pathological stage II or more advanced cancer, both OS and DSS were significantly worse in the BC changes group (50% vs. 81%, *p* < 0.001; 64% vs. 86%, *p* = 0.004, respectively) (Figure 5c,d).

### 3.5. Comparison of Cause of Death Between the BC Changes Group and the Non-Changes Group

Main causes of death are presented in Table 3. Deceased cases were observed in 85 patients. Gastric cancer recurrence was the most frequent cause of death in both groups (*n* = 44). Among the deaths from other diseases, malignant neoplasm of other organs (*n* = 8), cerebrovascular disease (*n* = 4), and infectious disease (*n* = 4) were observed. Infectious diseases included peritonitis and bacterial or viral pneumonia without aspiration. Postgastrectomy emaciation (PGE) was the second most frequent cause of death in the BC changes group. In this study, PGE death was defined as a postoperative disability due to progression of poor general condition that was derived from reduced oral intake, loss of vitality, malnutrition, decreased ADL, and/or subsequent disuse syndrome without cancer recurrence or any other obvious organic diseases causing death. When focusing on PGE death, it was significantly more prevalent in the BC changes group (35% vs 10%, *p* = 0.029).

### 3.6. Risk Factor Analysis for the BC Changes Group

Table 4 shows the results of risk factor analysis for the BC changes group. In the multivariate analysis, preoperative lower muscle density (OR, 1.882 [1.171–3.024], *p* = 0.009), higher Charlson comorbidity index (OR, 6.452 [1.346–30.939], *p* = 0.020), and total gastrectomy (OR, 3.315 [1.842–5.966], *p* < 0.001) were identified as significant independent risk factors. Meanwhile, in the univariate analysis, older age (*p* = 0.002), higher preoperative neutrophil lymphocyte ratio (NLR) (*p* = 0.044), more intraoperative blood loss (*p* < 0.001), and advanced pathological stage (*p* = 0.017) were significant factors.

## 4. Discussion

This study focused on postoperative gastric cancer patients, examining comprehensive changes in BC following surgery and evaluating long-term prognosis using a risk scoring method. The risk scores based on postoperative BC changes predicted the prognosis of postoperative patients with gastric cancer well.

After gastrectomy, patients frequently suffer from body weight loss due to decreased oral intake and digestive capacity. Human BC is approximately 40% skeletal muscle, 22% adipose tissue, 8% blood, 7% bone, and less than 4% skin, organs, and nervous system [18]. Because blood volume, bone, organs, and the nervous system are considered to be almost unchanged six months after surgery, the main cause of weight loss is presumably a decrease in skeletal muscle and adipose tissue, which account for approximately 60% of the human body. As in previous reports [7,12,13], we used available data at six months after gastrectomy in this study because that is when the first follow-up CT scan was during postoperative surveillance and no additional interventions just for research were required, and in advanced cases, occult cancer recurrence may be more influential on BC change at one year or more postoperatively. Recently, postoperative adjuvant chemotherapy has been frequently implemented for advanced cancer within 6 months after surgery. Postoperative chemotherapy was postulated to affect BC change; however, in this study population, the presence or absence of postoperative adjuvant chemotherapy was not identified as a significant risk factor for BC change, nor was the treatment period.

Muscle density was also included in this study to observe muscle quality. Intramuscular adipose tissue content (IMAC) has been reported to be one parameter of skeletal muscle quality. IMAC is a negative value expressed as a CT density ratio and is calculated by the following formula:IMAC = [CT density of the multifidus muscle]/[CT density of the dorsal subcutaneous fat]

When myosteatosis arises due to intramuscular fat deposition, muscle CT density decreases [19] and IMAC increases, approaching 0. As a result, high IMAC, which indicates myosteatosis, is associated with poor prognosis [20,21]. However, when calculating IMAC, regions of interest need to be selected at arbitrary tissue points, which may cause variations in the measurements, and because it is a minus-ratio parameter that is rather difficult to understand intuitively, we employed a simple evaluation method using the absolute CT value of the whole muscle. It should be noted that we do not deny IMAC.

Previous studies reported that GC patients with skeletal muscle loss of 5–19% at 6 to 12 months postoperatively had a poor prognosis [12,13,14]. Other studies have highlighted the association between comprehensive BC parameters and the prognosis of cancer patients. Decreases in skeletal muscle volume and density have been reported to worsen the prognosis in patients with liver cancer [4] and cancer cachexia [22]. However, this study differs from their report in that we focused on patients who have not yet developed cancer cachexia and have undergone surgical resection, with an emphasis on postoperative comprehensive changes in BC. Park et al. [7] classified BC into skeletal muscle volume, visceral fat volume, and subcutaneous fat volume separately, and reported that postoperative gastric cancer patients who experienced loss of one or more of these factors had a poor prognosis, which is generally consistent with our results. In contrast, our study differs from their report in that we evaluated body fat collectively, while skeletal muscle was assessed separately for both quantity and quality, with a double-weighted score assigned to muscle quantity. We believe that by emphasizing changes in skeletal muscle volume, given its strong impact on prognosis, we observed a gradual worsening of prognosis with an increase in the score, unlike their results.

We considered that the results of these BC parameter changes indicate the following. First, skeletal muscle volume, which had the highest hazard ratio compared with the other parameters, was regarded as the most powerful prognosticator, demonstrating that postoperative myopenia also has the strongest impact on survival, as reported in the preoperative period of GC patients [8]. Sarcopenia and systemic inflammation are known to be interrelated [23]. Previous studies have reported that individuals with sarcopenia tend to have higher C-reactive protein (CRP) levels and a higher NLR compared to the general population [24]. In the multivariate analysis of risk factors for BC changes in this study, inflammatory markers such as preoperative albumin, CRP, and NLR were not identified as significant factors. However, NLR was identified as a significant risk factor in the univariate analysis and may be a potential predictive factor for BC changes. We believe that additional investigation into the association between pre- and postoperative inflammatory marker changes and alterations in BC is crucial.

Second, regarding body fat, a reduction in body fat was associated with a poor prognosis in this study. Adipose tissue is a storehouse of reserved energy in the human body and is also implicated in immunological responses, promoting the activation and proliferation of immune cells [25]. Recent reports suggest that patients with a higher BMI have better survival from severe diseases such as heart failure, chronic obstructive pulmonary disease, renal failure, and sepsis, in a phenomenon known as the obesity paradox [26,27,28,29]. In patients with GC, preoperative decreases in body fat improves short-term prognosis [30,31]. Conversely, postoperative decreases in body fat worsened long-term prognosis in this study. Postoperative non-decrease in body fat may contribute to maintaining body energy, preserving immune activity, and resisting physical exhaustion, which may be related to better long-term prognosis. We speculated that the obesity paradox may also occur in postoperative patients with GC.

Finally, with respect to skeletal muscle density, myosteatosis has been regarded as a deterioration of skeletal muscle quality. It is caused by ectopic fat deposition within muscle tissue when the lipid volume exceeds the disposal capacity of muscle and adipose tissue and has been reported to be more prevalent with aging [32]. The clinical significance of sarcopenia has been widely discussed [33]; meanwhile, recent advances in sarcopenia research have led to the recognition of myosteatosis as a distinct disease separated from sarcopenia [34]. Myosteatosis disrupts metabolism, contributing to conditions such as insulin resistance and diabetes, which synergize with sarcopenia [35]. The present study showed an increase in skeletal muscle density after gastrectomy, a condition opposite to myopenia, which refutes previous post-gastrectomy reports [7]. When limited to surgical patients, the details of postoperative changes in skeletal muscle density in gastric cancer have not been elucidated. Therefore, the exact reason remains unclear; however, we propose the following mechanism. Postoperative increases in skeletal muscle density in this study were presumed to be caused predominantly by intramuscular defatting. We consider that this may be attributed to intramuscular fat catabolism as well as that of body fat tissue, resulting in a link between body fat consumption and increased muscle density. Hence, a weak negative correlation might be observed between body fat volume and skeletal muscle density (*r* = −0.430). Additionally, a decrease in intramuscular interstitial tissue components such as collagen fiber and water content, which have lower CT density than muscle tissue, may have also contributed to increase skeletal muscle density. Meanwhile, interestingly, there was no correlation between muscle volume and muscle density (*r* = 0.110). Changes in muscle volume and muscle density were equally indicative of muscle deterioration. Consequently, significant changes in both parameters after gastric cancer surgery were associated with sarcopenia. Thus, it was initially speculated that there might be an association between changes in muscle quantity and quality. However, the actual measurements showed no relationship between the two parameters. This may be attributed to various background factors, such as long-term muscle changes in the general population compared to the those observed during the six-month recovery period after gastric cancer surgery. We speculate that the lack of correlation between muscle density and muscle mass may be due to hypertrophy of degenerated muscle tissue compensating for the loss of fat, collagen fibers, and water content. Although assessing true muscle changes using living human tissue is difficult, histological imaging studies with other modalities, such as magnetic resonance imaging (MRI) and ultrasound, may yield valuable results in the future.

In a comparison of survival by cancer progression, the BC changes group of stage I cancers only showed a lower OS, whereas in advanced stage cases they exhibited both lower OS and DSS. This implies that in stage I patients, the number of deaths from non-cancerous causes was increased in the BC changes group, and that in advanced cancer cases, the incidence of BC changes was increased by the consequences of potential cancer progression as well as by non-cancerous death. As for cause of death, significantly more PGE deaths were observed in the BC changes group. Patients after gastrectomy often suffer from nutritional disorders and weight loss due to decreased food intake, reflux esophagitis, dumping symptoms, and impaired digestive and absorption capacity, which have been collectively known as postgastrectomy syndrome [36,37]. In this study, we defined PGE as the deterioration of general condition that occurs as a long-term consequence of these postgastrectomy disabilities. Postoperative changes in BC were considered to induce PGE death.

Regarding the clinical application of the results of this study, postoperative home interventions may be required to ensure that both muscle and body fat are unchanged as much as possible. Many reports mentioned the importance of preoperative intervention to improve BC in GC patients [5,8,9,10,11]; however, we are more concerned with postoperative maintenance and control of BC changes. We consider that cancer treatment should be prioritized in the preoperative period because a cancer-bearing status may be a cause of alternations in BC, and that it is unclear how much time should be spent on BC improvement. In contrast, postoperative intervention has the advantages of being free from the cancer-bearing condition and a non-urgent situation, has less mental strain on the patient by removing the cancer, and has a sufficient period of six months for intervention. BC changes have been shown to occur in the early period after gastrectomy, with a decrease of more than 5% in skeletal muscle and 10% in body fat at one month postoperatively [3,38]. Careful monitoring of postoperative BC changes may be required to prevent deterioration of BC for the first six months after gastrectomy. Post-discharge home nutrition for a specified period may also be beneficial [39], while further evidences have yet to be established to determine the indications and methods of nutritional intervention to prevent BC changes.

Furthermore, the results of this study identified risk factors for BC changes, including high comorbidity, preoperative myosteatosis, as well as total gastrectomy. For these high-risk patients, proactive surgical interventions may be required, such as avoiding total gastrectomy or adding a catheter-jejunostomy after total gastrectomy, if possible. Further studies are warranted to investigate the clinical efficacy of surgical and postoperative intervention. Although surgeons tend to focus mostly on curative resection and on postoperative complications, we may also have to pay attention to postoperative BC changes when making treatment decisions. De Felice et al. [40] reported that collaboration among multidisciplinary teams and the integration of insights from experts in oncology and nutrition could significantly enhance the quality of nutritional care and optimize patient outcomes. Additionally, the standardization of nutritional protocols has the potential to improve the survival rates of oncology patients. Periodic postoperative surveys using ADL assessment questionnaires may also be beneficial. Especially for high-risk patients with postoperative BC changes, strong nutritional management through a multidisciplinary oncology team—including nutritionists, physical therapists, and pharmacists, in addition to oncologists—may be essential for restoring the patient’s postoperative lifestyle to its preoperative status [41].

This study contains several limitations. This study is a retrospective study conducted at a single center. The regimen of postoperative adjuvant chemotherapy was based on the available evidence at the time but was not uniform. Post-discharge physical exercise was at the discretion of the patients, and their exercise levels were not standardized. Additionally, the validity of the BC cutoffs has not been verified. As previously reported [12,42,43], we determined the optimal cutoff value by ROC analysis for survival status and investigated the association between BC changes and long-term prognosis using the risk scoring method, which should be reevaluated by the external validation cohort. The last limitation is the adequacy of substituting the psoas muscle for skeletal muscle assessment. While conducting routine clinical practice, measuring total skeletal muscle on axial CT images requires dedicated analysis software and takes some effort. However, measuring the psoas muscle offers the advantage of being simple and convenient on many electronic medical record systems. Recently, psoas muscle index (PMI) has been reported to correlate with smooth muscle index (SMI) and has been shown to be one of the skeletal muscle parameters [44,45,46]. Meanwhile, other documents argue that the size of the psoas muscle is not a reliable predictor of sarcopenia due to individual variations and recommend further investigations to verify the use of PMI [47,48]. It may take additional time to establish the clinical utility of PMI [34]. However, at present, we believe that comparisons between patients are fully possible if all patients are evaluated using the same method including PMI. Recently, albumin-myosteatosis gauge (AMG) has been proposed as a novel skeletal muscle-related marker in colorectal and pancreatic cancer [49,50], with low pre-treatment AMG levels reported to worsen prognosis. AMG, expressed as [serum albumin level] × [skeletal muscle concentration], is a parameter that integrates nutritional status, systemic inflammation, and concomitant skeletal muscle wasting. It has gained interest as an indicator to assess the synergistic detrimental effects of these factors on patient survival. It would be very interesting to evaluate the changes in AMG before and after treatment and explore its potential role in gastric cancer patients, while it was not possible to evaluate AMG in this study. We consider this to be one of the topics to be examined in the future.

## 5. Conclusions

The present study demonstrated that each of postoperative skeletal muscle loss, body fat loss, and muscle hyperdensity negatively affected prognosis of GC patients after surgery, and the three-factor risk score we have proposed clearly predicted the postoperative prognosis of patients with GC. We believe that this study provides new insights into a nutritional imaging biomarker.

## Figures and Tables

**Figure 1 cancers-17-00738-f001:**
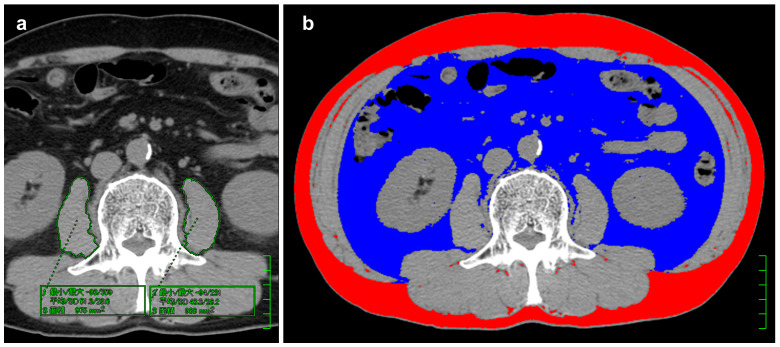
Cross-sectional plane computed tomography (CT) images at the third lumbar vertebra. (**a**) The right and the left psoas major muscles were traced semi-automatically by placing the region of interest; the total muscle volume and the total CT density were defined as the muscle volume and CT density. The green boxes represent the measured values of the traced regions of the right and left psoas muscles. The top row indicates the maximum pixel density/minimum pixel density, the middle row shows the mean pixel density/standard deviation of pixel density, and the bottom row displays the calculated area of each region. (**b**) Body fat volume was measured as the sum of subcutaneous adipose tissue (red area) and visceral adipose tissue (blue area) using a commercially based software (Fat tissue analyzer version 3.0, Virtual Place Fujin Raijin; Aze, Canon Medical Systems, Tochigi, Japan) with tissue thresholds of −190 to −30 HU.

**Figure 2 cancers-17-00738-f002:**
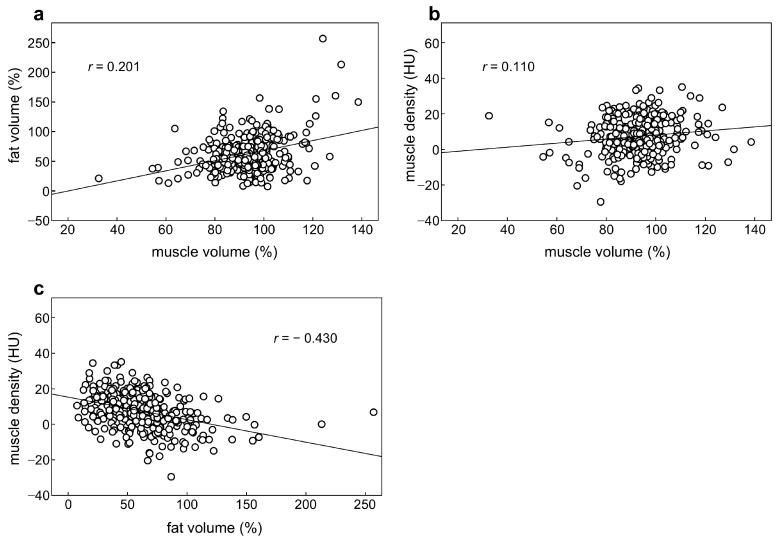
Correlations of body composition parameters changes. (**a**) Muscle volume changes and fat volume changes, (**b**) muscle volume changes and muscle density changes, and (**c**) fat volume changes and muscle density changes. HU, Hounsfield unit.

**Figure 3 cancers-17-00738-f003:**
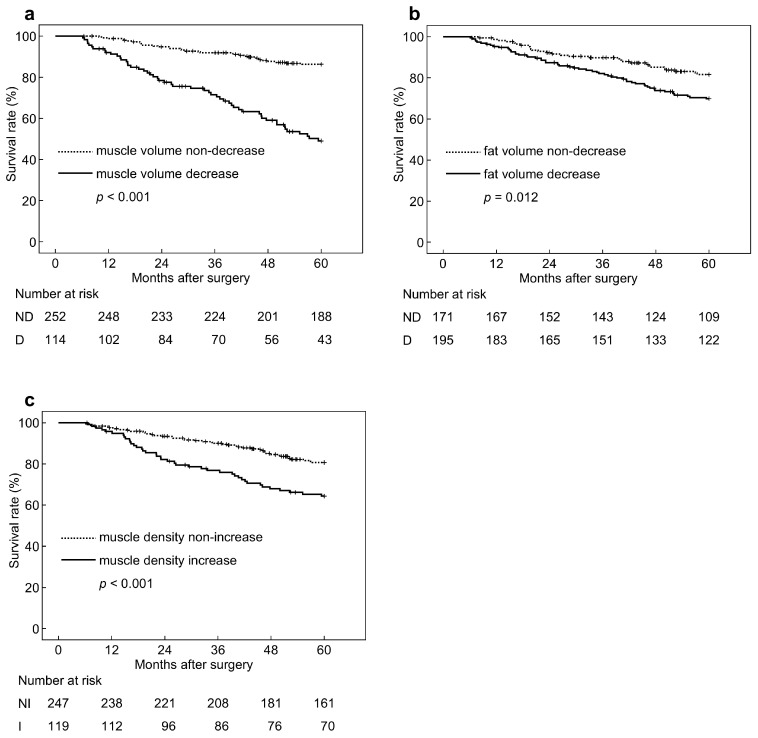
Overall survival curves according to body composition parameters. (**a**) Muscle volume change, (**b**) fat volume change, (**c**) muscle density change. ND, non-decrease; D, decrease; NI, non-increase; I, increase.

**Figure 4 cancers-17-00738-f004:**
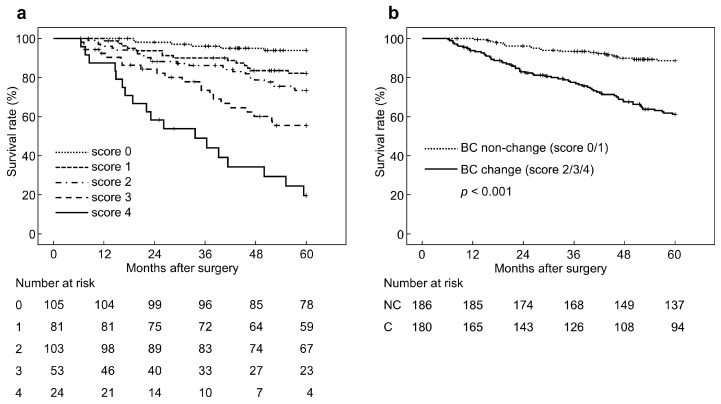
Overall survival curves according to the risk score of body composition changes. (**a**) Minor subgroups, (**b**) major subgroups. BC, body composition; NC, non-change; C, change.

**Figure 5 cancers-17-00738-f005:**
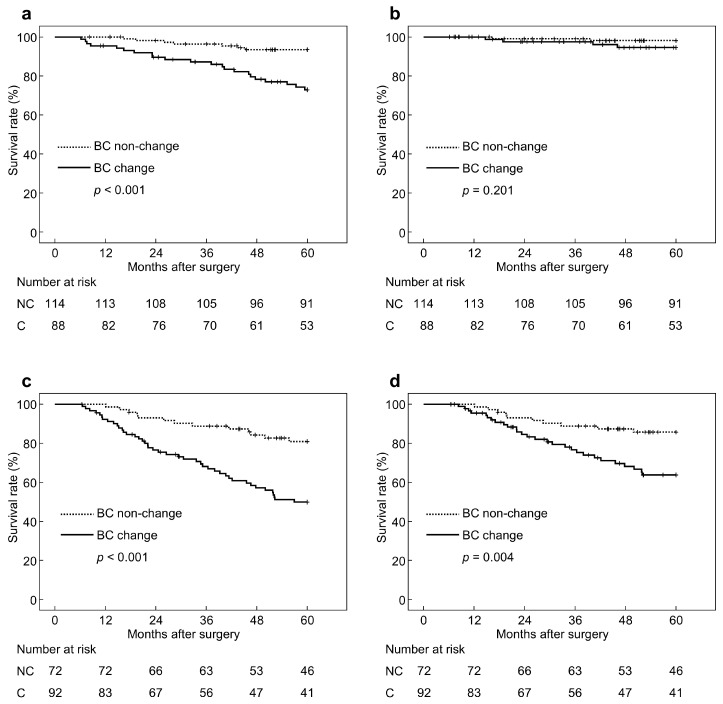
Overall survival (OS) and disease-specific survival (DSS) according to cancer progression. (**a**) OS of pathological stage I cancer, (**b**) DSS of pathological stage I cancer, (**c**) OS of pathological stage II or more advanced cancer, (**d**) DSS of pathological stage II or more advanced cancer. BC, body composition; NC, non-change; C, change.

**Table 1 cancers-17-00738-t001:** Patents characteristics.

Variables	
Median age, years	71 [64–76]
Sex	
male/female	258/108
BMI, kg/m^2^	22.3 [20.5–24.8]
Charlson comorbidity index	
0/1/2/≥3	265/30/58/13
Tumor location	
upper/middle/lower	82/151/133
Lauren classification	
intestinal/diffuse	199/167
Neoadjuvant chemotherapy	
yes/no	22/344
Surgical approach	
laparotomy/laparoscopy	187/179
Extent of gastrectomy	
total/distal/proximal	87/259/20
Extent of lymphadenectomy	
<D2/≥D2	219/147
Pathological stage	
I/II/III/IV	202/77/82/5
Postoperative adjuvant chemotherapy	
yes/no	153/213
Preoperative body composition	
Muscle volume, cm^2^/m^2^	5.05 [4.01–6.33]
Fat volume, cm^2^/m^2^	85.4 [54.5–116.2]
Muscle density, HU	81.7 [69.5–91.8]
Postoperative body composition change	
Muscle volume, %	94.2 [87.1–99.1]
Fat volume, %	60.6 [42.1–75.3]
Muscle density, HU	+7.1 [+1.8–+14.2]
Period of surgery divided by midpoint (June 2013)	
first term/second term	197/169

Continuous variables were expressed as median [Interquartile range]. BMI, body mass index; HU, Hounsfield unit.

**Table 2 cancers-17-00738-t002:** Prognostic analysis for survival.

		5y-OS	Univariate Analysis		Multivariate Analysis		
Prognostic Factor	*n*	(%)	HR [95% CI]	*p*-Value	Coefficient	HR [95% CI]	*p*-Value
Age							
≥76 years	104	60	2.470 (1.612–3.786)	<0.001	0.863	2.370 (1.506–3.729)	<0.001
<76 years	262	81	1				
Sex							
male	258	74	1.163 (0.715–1.889)	0.543			
female	108	78	1				
Preoperative BMI							
<20.5 kg/m^2^	90	66	1.722 (1.099–2.696)	0.018	0.466	1.594 (1.000–2.539)	0.050
≥20.5 kg/m^2^	276	78	1				
Preoperative Alb							
<4.2 g/dL	227	68	2.620 (1.557–4.410)	< 0.001	0.375	1.454 (0.825–2.564)	0.195
≥4.2 g/dL	139	87	1				
Preoperative TC							
<199 mg/dL	227	70	1.887 (1.161–3.066)	0.010	0.140	1.150 (0.683–1.937)	0.598
≥199 mg/dL	139	83	1				
Preoperative NLR							
≥2.54	127	69	1.531 (0.997–2.350)	0.052			
<2.54	239	79	1				
Charlson comorbidity index							
≥2	71	55	2.582 (1.654–4.031)	<0.001	0.587	1.799 (1.116–2.900)	0.016
≤1	295	80	1				
Lauren classification							
diffuse	167	73	1.293 (0.845–1.978)	0.237			
intestinal	199	78	1				
Neoadjuvant chemotherapy							
yes	22	76	1.576 (0.761–3.265)	0.220			
no	344	89	1				
Surgical approach							
laparotomy	187	66	2.571 (1.612–4.101)	<0.001	0.099	1.104 (0.604–2.020)	0.748
laparoscopy	179	85	1				
Extent of gastrectomy							
total	87	55	3.030 (1.975–4.648)	<0.001	0.603	1.828 (1.136–2.942)	0.013
distal/proximal	279	82	1				
Extent of lymphadenectomy							
≥D2	147	72	1.275 (0.831–1.955)	0.266			
<D2	219	78	1				
Intraoperative blood loss							
≥1.87 g/kg	206	67	2.742 (1.661–4.527)	<0.001	0.304	1.355 (0.718–2.556)	0.348
<1.87 g/kg	160	86	1				
Postoperative complication							
≥Grade 3	35	66	1.466 (0.778–2.763)	0.236			
≤Grade 2	331	76	1				
Pathological stage							
≥II	164	64	2.725 (1.740–4.268)	<0.001	0.866	2.377 (1.256–4.501)	0.008
I	202	85	1				
Postoperative adjuvant chemotherapy							
yes	153	70	1.561 (1.020–2.388)	0.040	−0.156	0.856 (0.480–1.527)	0.598
no	213	79	1				
Period of surgery							
first half	197	75	1.053 (0.684–1.620)	0.815			
second half	169	76	1				
BC change							
change (score ≥ 2)	180	61	4.049 (2.452–6.687)	<0.001	1.127	3.086 (1.831–5.202)	< 0.001
non-change (score ≤ 1)	186	89	1				

Cutoff values for continuous variables were determined by calculating the Youden index in the receiver operating characteristic analysis for survival. BMI, body mass index; BC, body composition; Alb, albumin; TC, total cholesterol; NLR, neutrophil lymphocyte ratio; OS, overall survival; HR, hazard ratio; CI, confidence interval.

**Table 3 cancers-17-00738-t003:** Comparison of cause of death between the BC changes group and the non-changes group.

Cause of Death	BC Changes Group, *n* (%)	BC Non-Changes Group, *n* (%)	*p*-Value
Main cause			0.123
gastric cancer recurrence	32 (50)	12 (60)	
malignant neoplasm of other organs	6 (9)	2 (10)	
cerebrocardiovascular disease	2 (3)	2 (10)	
infectious disease	2 (3)	2 (10)	
PGE	23 (35)	2 (10)	
Postgastrectomy emaciation			
yes	23 (35)	2 (10)	0.029
no	42 (65)	18 (90)	

**Table 4 cancers-17-00738-t004:** Risk factor analysis for body composition change.

	BC Change		Univariate Analysis		Multivariate Analysis		
Risk Factor	Yes, *n* (%)	No, *n* (%)	OR [95% CI]	*p*-Value	OR [95% CI]	*p*-Value	VIF
Age, year							
≥70 years	113 (63)	87 (47)	1.919 (1.264–2.919)	0.002	1.481 (0.926–2.367)	0.101	1.142
<70 years	67 (37)	99 (53)	1				
Sex							
female	57 (32)	51 (27)	1.227 (0.782–1.924)	0.373			
male	123 (68)	135 (73)	1				
ASA-PS							
3	23 (13)	16 (9)	1.557 (0.793–3.054)	0.196			
≤2	157 (87)	170 (91)	1				
Preoperative BMI							
≥22.1 kg/m^2^	104 (58)	91 (49)	1.429 (0.946–2.158)	0.090			
<22.1 kg/m^2^	76 (42)	95 (51)	1				
Preoperative muscle volume							
≥5.73 cm^2^/m^2^	71 (39)	65 (35)	1.213 (0.793–1.854)	0.373			
<5.73 cm^2^/m^2^	109 (61)	121 (65)	1				
Preoperative fat volume							
≥128.5 cm^2^/m^2^	38 (21)	27 (15)	1.576 (0.916–2.712)	0.099			
<128.5 cm^2^/m^2^	142 (79)	159 (85)	1				
Preoperative muscle density							
<82.7 HU	110 (61)	84 (45)	1.908 (1.259–2.893)	0.002	1.882 (1.171–3.024)	0.009	1.148
≥82.7 HU	70 (39)	102 (55)	1				
Preoperative Alb							
<4.0 g/dL	86 (48)	77 (41)	1.295 (0.857–1.958)	0.220			
≥4.0 g/dL	94 (52)	109 (59)	1				
Preoperative TC							
<202 mg/dL	126 (70)	118 (63)	1.345 (0.869–2.081)	0.183			
≥202 mg/dL	54 (30)	68 (37)	1				
Preoperative CRP							
≥0.95 mg/dL	21 (12)	15 (8)	1.506 (0.750–3.211)	0.247			
<0.95 mg/dL	159 (88)	171 (92)	1				
Preoperative NLR							
≥2.09	106 (59)	90 (48)	1.529 (1.011–2.309)	0.044	1.410 (0.903–2.201)	0.130	1.023
<2.09	74 (41)	96 (52)	1				
Charlson comorbidity index							
≥3	11 (6)	2 (1)	5.988 (1.308–27.048)	0.009	6.452 (1.346–30.939)	0.020	1.010
≤2	169 (94)	184 (99)	1				
Lauren classification							
diffuse	86 (48)	81 (44)	1.186 (0.786–1.790)	0.417			
intestinal	94 (52)	105 (56)	1				
Neoadjuvant chemotherapy							
yes	12 (7)	10 (5)	1.257 (0.529–2.987)	0.604			
no	168 (93)	176 (95)	1				
Surgical approach							
laparotomy	97 (54)	90 (48)	1.247 (0.827–1.879)	0.292			
laparoscopy	83 (46)	96 (52)	1				
Extent of gastrectomy							
total	62 (34)	25 (13)	3.384 (2.009–5.700)	< 0.001	3.315 (1.842–5.966)	< 0.001	1.227
distal/proximal	118 (66)	161 (87)	1				
Extent of lymphadenectomy							
≥D2	76 (42)	71 (38)	1.184 (0.779–1.798)	0.429			
<D2	104 (58)	115 (62)	1				
Intraoperative blood loss							
≥4.01 g/kg	86 (48)	57 (31)	2.017 (1.350–3.175)	< 0.001	1.239 (0.745–2.061)	0.410	1.308
<4.01 g/kg	94 (52)	129 (69)	1				
Postoperative complication							
≥Grade 2	65 (36)	56 (30)	1.312 (0.848–2.030)	0.222			
≤Grade 1	115 (64)	130 (70)	1				
Pathological stage							
>II	92 (51)	72 (39)	1.655 (1.093–2.507)	0.017	1.249 (0.781–1.997)	0.354	1.141
I	88 (49)	114 (61)	1				
Postoperative adjuvant chemotherapy							
yes	80 (44)	73 (39)	1.238 (0.817–1.877)	0.314			
no	100 (56)	113 (61)	1				
Period of surgery							
first term	100 (56)	97 (52)	1.147 (0.760–1.730)	0.514			
second term	80 (44)	89 (48)	1				

Categorical variables were classified into two groups according to the minimum *p*-value in the chi-square test, and continuous variables were divided by calculating the Youden index from the receiver operating characteristic analysis. BMI, body mass index; ASA-PS, American Society of Anesthesiologists Physical Status; Alb, albumin; TC, total cholesterol; CRP, C-reactive protein; NLR, neutrophil lymphocyte ratio; BC, body composition; OR, odds ratio; CI, confidence interval; VIF, variance inflation factor.

## Data Availability

The datasets generated and/or analyzed during the current study are available from the corresponding author on reasonable request.

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
