# Peer review of "Postoperative Changes in Body Composition Predict Long-Term Prognosis in Patients with Gastric Cancer"

_cancers, 2025, doi:10.3390/cancers17050738_

Round 1
Reviewer 1 Report
Comments and Suggestions for Authors
The study investigates the prognostic impact of postoperative changes in body composition (BC) in gastric cancer (GC) patients undergoing radical gastrectomy. The authors present a comprehensive analysis of skeletal muscle and fat volume changes, incorporating a risk scoring system.I would like to offer the following points for consideration by the authors towards the improvement of the manuscript.
1- The authors claim this is the first study evaluating postoperative BC changes in a comprehensive manner. However, similar studies on postoperative sarcopenia and body fat changes exist. The authors should more explicitly differentiate their work from prior research.
2- Several potential confounders such as chemotherapy regimens, baseline nutritional status, and exercise levels are not thoroughly addressed. Were these factors controlled for in the statistical analysis?
3- The use of psoas muscle index (PMI) as a surrogate for total skeletal muscle mass is debated. The authors should either justify its reliability or supplement it with additional muscle indices.
4- The authors suggest that body fat loss correlates with worse outcomes, possibly due to reduced energy stores. However, other studies have shown contrasting results in different oncologic settings. Can the authors discuss alternative explanations?
5- The study should discuss the potential role of the albumin-myosteatosis gauge as a novel marker in assessing postoperative nutritional and prognostic status. Recent studies suggest that integrating albumin levels with myosteatosis assessment could enhance predictive accuracy.
Reviewer 2 Report
Comments and Suggestions for Authors
In this study, the authors investigated the impact of postoperative changes in body composition (BC), such as skeletal muscle volume, body fat volume, and skeletal muscle density, to the prognosis of postoperative patients with gastric cancer (GC). The overall study is clear, however, they need further clarification for the following points:
1. The authors demonstrated the importance of changes in BC, specifically in quantity and quality of skeletal muscle, on the prognosis of patients with GC. However, considering the significance of sarcopenia has been widely discussed (PMID:36586787), is there a relationship between the changes in quantity and quality of skeletal muscle and sarcopenia? Are these changes independent of sarcopenia? The authors should discuss the relationship between the changes in quantity and quality of skeletal muscle and sarcopenia within the manuscript.
1. It has been reported that inflammatory index are crucial for the levels of skeletal muscle and are also independent prognostic factors for patients with malignancy. Do perioperative changes in inflammatory index correlate with the alterations in skeletal muscle, or with the changes in BC?
3.The authors illustrated that the extent of gastrectomy is a significant factor for GC. When incorporating both the extent of gastrectomy and changes in GC into multivariate analysis for prognosis, should attention be paid to the issue of multicollinearity?
Reviewer 3 Report
Comments and Suggestions for Authors
Our research colleagues have produced this paper whose aim is to study postoperative skeletal muscle loss, body fat loss and muscle hyperdensity that negatively affect the prognosis of gastric cancer patients after surgery. Excellent introduction in which the nutritional status of the patient affected by gastric neoplasia is greatly affected. We must now think about the possibility of feeding these patients with professionals who are part of the oncology team and who deal with this matter (doi.org/10.3390/nu17010188 to be read and cited in the bibliography). Excellent description in the materials and methods that also exposes the neoadjuvant therapy that was, we imagine, discussed in a multidisciplinary commission, for each patient. Effective to study fat and muscle mass with CT, given that this test is repeated several times from the time of initial staging to the various follow-up steps. It would be useful to make a comparison with the results of impedance analysis. Excellent the scale determined by the score assigned based on the result of the CT scan. Discussion on which we agree, excellent the sentence "It has been hypothesized that postoperative chemotherapy influences the change in body composition, however, in this study population, the presence or absence of postoperative adjuvant chemotherapy has not been identified as a significant risk factor for the change in BC, nor was the treatment period. In any case, it can be recommended to always evaluate with CT if there are changes in fat and lean mass during adjuvant chemotherapy. We partially agree on changing the surgical approach because the outcome is not to be curative while I find it more correct to address these problems with a nutritionist. Maybe you can also suggest administering a questionnaire on the quality of life to the patient that could also give us additional indications (PMID: 9617108 to read and cite in the bibliography). We agree on the limitations and conclusions. Good iconography, good English, good bibliography
Round 2
Reviewer 1 Report
Comments and Suggestions for Authors
I am satisfied that the authors have addressed all of my previous concerns about the article. It is now much improved and I feel that it is now suitable for publication.